# HiS4MAE: High-efficiency Segmentation of Subcellular Structure via Self-distillated Masked Autoencoder

## Abstract

The accurate identification of subcellular structures is crucial for understanding cellular functions. However, due to the varied morphology of different cells, conventional segmentation methods typically depend on a substantial collection of accurately labeled images of cell structures. The creation of such precise labels is often time-consuming and labor-intensive. To address this issue, we introduce an efficient, self-supervised method for segmenting subcellular structures, named HiS4MAE (High-efficiency Segmentation of Subcellular Structure via Self-distillated Masked Autoencoder). Leveraging an enhanced masked autoencoder (MAE), we train the encoder using the masked image modeling (MIM) framework, followed by clustering the encoded high-dimensional features to achieve pixel-level segmentation of structures. We employ a self-distillation technique to accelerate the model's training process and propose an inference method that is less time-consuming. We also introduce a discrete codebook to assist the self-distillation process, enhancing the model's stability during training. When applied to a publicly available volumetric electron microscopy (VEM) dataset of primary mouse pancreatic islet $\beta$ cells, HiS4MAE not only surpasses the state-of-the-art technique but also significantly reduces the time required for both training and inference.

## 1 Introduction

Biologists use microscopes to understand cells. In this process, segmenting the various components within cells (such as nuclei, mitochondria, and Golgi apparatus) from microscopy images is crucial. This technique is known as subcellular structure segmentation. Precisely segmenting subcellular structures at the pixel level helps researchers understand cellular functions and simulate complex biological processes within cells. With the continuous advancement of microscopy imaging technology, acquiring images is no longer a bottleneck. Techniques such as volumetric electron microscopy and confocal laser scanning microscopy (Peddie et al., 2022; Amos & White, 2003) can now capture hundreds of gigabytes to terabytes of data at speeds ranging from several frames to hundreds of frames per second. This allows researchers to easily observe the fine structures within cells, but how to efficiently utilize these massive amounts of images is a problem worth considering.

Some supervised image segmentation methods, such as those based on the convolutional neural network or the transformer (Dosovitskiy et al., 2021), have achieved remarkable results. However, these methods often require a large number of precisely annotated images for training to achieve satisfactory performance. Annotating different types of images requires domain experts to invest significant time and effort. In reality, it is impractical for a few experts to manually annotate hundreds of gigabytes to terabytes of images, which greatly limits the utilization of these images by researchers.

Consequently, self-supervised or unsupervised methods have gained the attention of researchers. Notably, the same organelles tend to exhibit similar textural features in images. Research (Hsu et al., 2021) has shown that by using representation learning to distinguish these textures, different organelle regions can be segmented. Han et al. used a variational autoencoder (VAE) (Kingma & Welling, 2013) to encode image patches, mapping textured image patches to a high-dimensional

space for clustering, achieving relatively good results and providing a new approach for self-supervised segmentation (Han et al., 2022).

This method of encoding image patches is reminiscent of MAE, a masked image modeling (MIM) method that trains the model to recover masked image patches, thereby learning deep features of the images. Xie et al. proposed a self-supervised segmentation method (Xie et al., 2023) based on MAE, using a technique called cover-stride to expand encoded features into the latent space and then performing clustering in the feature space to achieve pixel-level segmentation. This method is effective but still has some issues. The performance of K-means clustering depends on the selection of initial cluster points, and different cluster points can result in significantly different clustering outcomes. Additionally, while cover-stride is an effective method, its inference requires multiple model invocations, resulting in considerable time consumption. Chen et al. indicate that a self-distillation architecture can significantly increase the training efficiency of MIM methods. They propose an architecture named the self-distilled masked autoencoder (SdAE) (Chen et al., 2022). This architecture surpasses the performance of MAE after 1600 training epochs with only 300 training epochs. However, this method's pretext task differs from MAE, leading to unstable performance during training and some degree of degradation.

In this work, we propose a novel approach to address these issues. We hypothesize that MAE outperforms VAE because the MIM framework allows the model to learn more high-level features. However, simple decoding methods like cover-stride can make it difficult to cluster overly abstract features, leading to segmentation failure. The key is to balance these factors. We incorporated the self-distillation architecture from SdAE but retained the MAE pretext task of reconstructing pixels. Additionally, we used a discrete codebook to assist the model's self-distillation, preventing degradation during training. Our subsequent experiments demonstrated the effectiveness of this approach. To eliminate the impact of different initial cluster points on segmentation results, we selected fixed cluster centers. Furthermore, we developed a method called expand-stride to reduce the time cost of inference. Experiments have shown that our method is highly effective. In summary, our contributions include:

- We introduce self-distillation to enhance the performance of the encoder, improving segmentation effects while reducing training costs. Our method only requires training for 700 epochs to surpass the effects achieved by the original method after 2800 epochs. At the same time, it increases the overall segmentation effect by more than 13%.
- To our knowledge, we are the first to use the discrete codebook to assist in model self-distillation. This method effectively prevents model degradation during the training process and further enhances the model's efficiency in utilizing the dataset.
- We propose an inference method named expand-stride, which significantly reduces the time complexity during inference compared to the cover-stride method in the state-of-the-art approach.

## 2 RELATED WORK

### 2.1 APPLICATION OF DEEP LEARNING TO MICROSCOPIC IMAGE

With the advancement of deep learning, its impact on the field of microscopic imaging has become increasingly significant. Numerous pivotal models proposed in the realm of deep learning have been extensively applied in the domain of biological imaging. Since its inception, U-net (Ronneberger et al., 2015) has emerged as a vital paradigm in the field of biological imaging. Works such as RepMode (Zhou et al., 2023) and Sparse SSP (Zheng et al., 2024) have built upon U-net to achieve precise predictions of subcellular structures in microscopic images, yet similar approaches generally necessitate authentic and accurate labels for training. For unsupervised or self-supervised methods, VQ-VAE (Van Den Oord et al., 2017), VAE (Kingma & Welling, 2013), and ViT (Dosovitskiy et al., 2021) have also played crucial roles. A viable paradigm involves training a model with some pretext task, and subsequently using only the encoder to process images, followed by clustering of the encoded features to yield results for downstream tasks. Han et al. leverage VAE and metric learning to fit the distribution of image patches in latent space, ultimately obtaining segmentation outcomes through K-means. Similar methodologies are not confined to segmentation tasks; for instance, cytoself (Kobayashi et al., 2022), based on VQ-VAE, also accomplishes protein localization

analysis and clustering at the subcellular level through the pretext task of image reconstruction. This demonstrates the formidable feature extraction capabilities of neural networks and substantiates the immense potential of self-supervised approaches in the field of microscopic imaging.

## 2.2 SELF-SUPERVISED LEARNING AND MASKED IMAGE MODELING

Han et al. contend that this paradigm is more effective when the global context of the image is not critical for downstream tasks. In contrast, the researchers behind MAESTER (Xie et al., 2023) propose an alternative approach. Rather than employing convolutional neural networks as encoders, they transitioned to a masked autoencoder (MAE) based on transformer architecture, resulting in superior performance. Transformers are widely regarded as more adept at capturing long-range global dependencies compared to CNNs. Consequently, they posit that this ability to capture global dependencies facilitates the segmentation of subcellular structures.

Numerous studies, such as CAE (Chen et al., 2024) and SdAE (Chen et al., 2022), focus on optimizing the encoder of MAE. CAE introduces a regressor with cross-attention to further decouple the encoder and decoder, enhancing the performance of both. However, this also results in more abstract features post-encoding, which are not conducive to subsequent segmentation tasks. SdAE, building upon the foundation of MAE, incorporates an encoder without gradients as a teacher branch and modifies the pretext task of MAE, transforming the decoding target into a vector with the same dimensionality as the latent space. Experimental results indicate that this approach indeed improves the training efficiency of the encoder, but in terms of the downstream task of segmentation, the model's performance post-clustering after changing the pretext task is not satisfactory.

## 2.3 DISCRETE CODEBOOK

The mechanism of the codebook has been employed since the advent of VQ-VAE, encoding images into a discrete matrix to reduce the time complexity of generating images. Such a codebook effectively serves as a cluster center during the training process. The subsequent BEiT (Bao et al., 2021) also introduced this mechanism, utilizing dVAE to generate discrete encodings. Recent researchers (Du et al., 2024) theoretically substantiate the enhancement of the MIM framework by the codebook mechanism. Choosing a codebook that is more aligned with data classes can improve the model's performance.

## 3 METHOD

### 3.1 SUBCELLULAR STRUCTURE SEGMENTATION AND MASKED IMAGE MODELING

Subcellular structure segmentation within microscopic images is a specialized task that necessitates the delineation of various cellular organelles. To formalize this, let $X$ denote a collection of 3-dimensional microscopic cell images, each with dimensions $D \times H \times W$. Correspondingly, $Y$ represents a set of fully segmented subcellular structure images, also dimensioned $D \times H \times W$, where each pixel in $Y$ is labeled from 0 to $T - 1$, with $T$ representing the total number of distinct cell types to be segmented. The objective is to develop a mapping function $f$, implemented via a neural network, that accurately transforms $X$ into $Y$, i.e., $f : X \to Y$.

Drawing inspiration from the work of MAESTER (Xie et al., 2023), we adopt the masked image modeling (MIM) framework to address this challenge. Given that the dimensions of the images in $X$ and $Y$ are substantial, on the order of $10^3$ in each spatial dimension, it is impractical to process the entire image in one pass through the network. Consequently, we initiate the process by randomly sampling a 2D slice $X_i$ from $X$. Following the terminology introduced by MAESTER, we refer to the shape of $X_i$ as the field of view (FOV), with our FOV matching that of Xie et al., specifically $80 \times 80$ pixels.

In the context of MIM approaches, the process involves segmenting the 2D image $X_i$ into $n$ patches of uniform size, where $n$ is the count of perfect squares. Subsequently, a set of binary masks $m$, with $m_i \in \{0, 1\}$ for $i = 0, 1, 2, \ldots, n - 1$, is used to partition $X_i$ into two disjoint subsets, $X_i^{'}$ ($m_i = 0$) and $\overline{X_i^{'}}$ ($m_i = 1$). The core objective of the MIM approach is then to minimize a specific loss

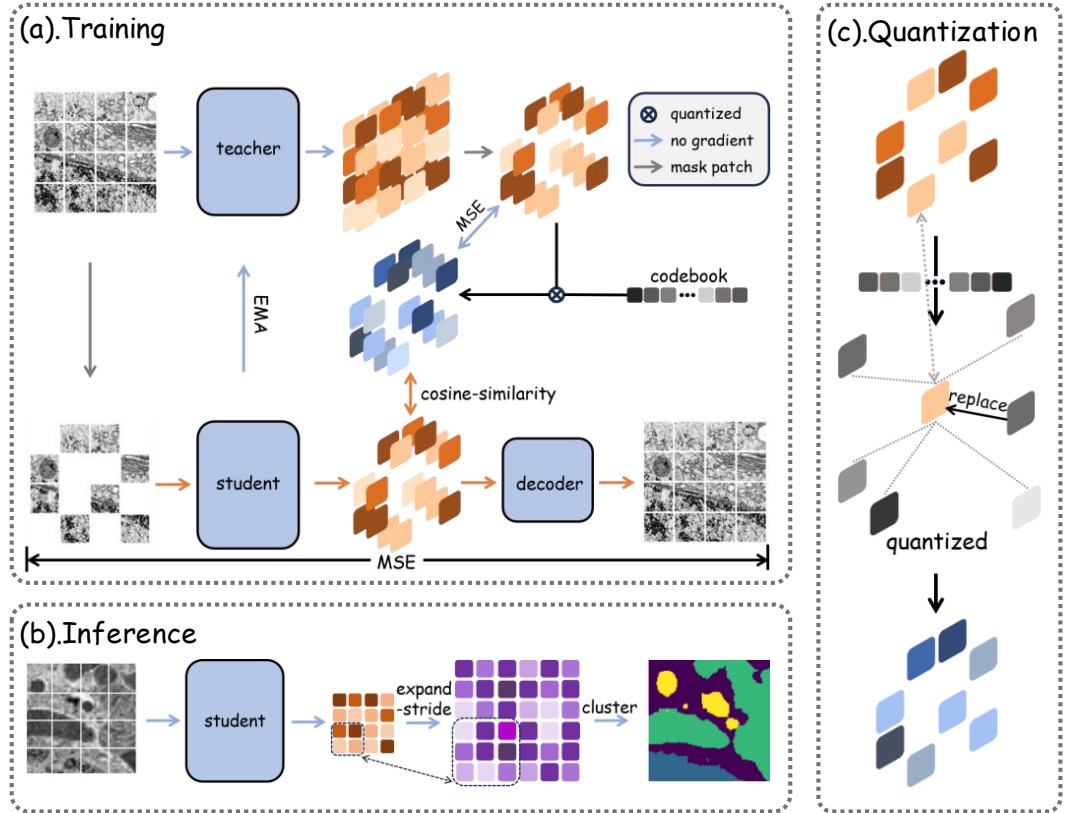

Figure 1: **(a) The overall structure of the HiS4MAE.** Its pretext task is to reconstruct the masked image patches. **(b) The process of cell segmentation.** It only uses the encoder of the student branch to infer the complete image, and after clustering the inferred high-dimensional features, segmentation is completed. **(c) The mechanism of the codebook.** For each high-dimensional vector, select the one closest in the codebook's vectors to replace it.

function, which we denote as:

$$Loss = \frac{1}{n} \sum_{i=0}^{n-1} L(h(X_i^{'}), t(\overline{X_i^{'}})) \tag{1}$$

Here, $h(x)$ is a function that reconstructs the original image from the unmasked $X_i^{'}$, while $t(x)$ serves as a transformation function, which in some architectures, represents the tokenizers. But in MAE (He et al., 2022), $t(x)$ is an identity function, in other words, $t(x) = x$. The exact forms of $h(x)$ and $t(x)$ are subject to variation depending on the specific neural network architecture employed.

In our approach, aligning with the methodology of MAE, the loss function $L$ is defined as the mean square loss (MSE). This choice is made to ensure robustness and accuracy in the segmentation of subcellular structures across diverse cell types.

### 3.2 MODEL ARCHITECTURE

Since the inception of convolutional neural networks (CNN), they have achieved remarkable success across various domains of deep learning, establishing themselves as a dominant approach in computer vision. However, with the advent of Transformer in the field, there is a growing consensus among researchers that the attention mechanism employed by transformers excels at capturing global image information compared to CNN. Our methodology leverages MAE and integrates attention blocks within an encoder-decoder framework. Empirical evidence from numerous experiments

suggests that asymmetric encoder-decoder configurations enhance the model's focus on the encoding process. Building upon the ViT-B foundation, MAESTER has further optimized the parameter count. Our baseline model architecture settings are derived from MAESTER.

Motivated by the SdAE (Chen et al., 2022), we have incorporated a self-distillation structure into our model, which is anticipated to significantly curtail the training duration. The architecture comprises two encoders: a teacher branch and a student branch, both sharing an identical model architecture. During training, the teacher network refrains from updating gradients, opting instead for an exponential moving average (EMA) approach to parameter updates. It is noteworthy that SdAE's primary objective is not the reconstruction of masked image patches but rather the alignment of features within the latent space. This contradicts our initial hypothesis that the reconstruction of the image itself aids in the task of subcellular structure segmentation. Our experiments have also demonstrated that it is challenging to achieve subcellular structure segmentation of microscopic cell images by training encoders solely through the alignment of features in the latent space. Consequently, our model retains the the same pretext task as utilized in MAE. Let the reconstructed image be $\hat{X}_i$, we show the loss function of the reconstruction as follow:

$$L_{recon} = MSE(X_i, \hat{X}_i) \tag{2}$$

Our initial idea was to augment the model of SdAE with an additional decoder for image reconstruction. In practical experimentation, we observed a non-linear correlation between model reconstruction efficacy and segmentation performance. In essence, the model's proficiency in reconstructing the original image does not necessarily translate to effective clustering of high-dimensional features. As training progresses, while the reconstruction loss remains relatively stable, the segmentation performance of the encoder tends to deteriorate. We will elaborate on this point in the subsequent experimental section. To address this issue, we have introduced a discrete codebook to facilitate model self-distillation. The final architecture of our model is depicted in Figure 1. The codebook is an embedding layer of shape $k \times embed\_dim$. We define the feature encoded by the encoder as $Z$ and the vector set within the codebook as $Z_q$. Our strategy aims to minimize the discrepancy between $Z_q$ and the teacher branch's encoding $Z_{tea}$, subsequently leveraging $Z_q$ to bolster the student branch's training. To this end, we employ MSE to constrain $Z_q$ to $Z_{tea}$ and cosine similarity to align $Z_q$ with $Z_{stu}$, expressed as:

$$L_{tea} = MSE(Z, Z_q) \tag{3}$$
$$L_{stu} = 1 - cosine\_similarity(Z_q, Z_{stu}) \tag{4}$$

In summary, our final loss function is formulated as:

$$Loss = L_{recon} + \alpha L_{tea} + \beta L_{stu} \tag{5}$$

Mechanisms analogous to the codebook have been previously proposed in the literature (Hirsch et al., 2023; Van Den Oord et al., 2017). In this context, we draw inspiration from the approach taken in VQ-VAE (Van Den Oord et al., 2017), setting $\beta = 0.25\alpha$. The values of these parameters are shown in Table 1

However, this approach may encounter a challenge akin to that faced by the deep cluster (Caron et al., 2018): trivial solutions may emerge. Specifically, the majority of $Z_{tea}$ are clustered near a small number of $Z_q$, leading to a reduction in the accuracy of image segmentation results. To counteract this, we automatically update the isolated part of $Z_q$ after each training epoch using the following adjustment:

$$Z_q^{isolated} = \gamma \times Z_q^{surrounded} + (1 - \gamma) \times Z_q^{isolated}, \ \ \gamma \in (0, 1) \tag{6}$$

where $\gamma$ is a hyperparameter that controls the update rate of $Z_q$.

### 3.3 MODEL INFERENCE

After the model training is completed, we only utilize the student branch of the model's encoders for inference. During the inference process, we no longer perform mask operations on the images; instead, we feed all the slices $X_i$ into the encoder, encoding $X_i$ into a matrix $Z_i$ of size $n \times embed\_dim$. As we know, the semantics of pixels within the same patch are roughly similar. Consequently, we employ a linear function to replace the nonlinear function to fit the value of each pixel within the patch, thereby obtaining features $Z_i$ that match the size of FOV. Subsequently, we infer the entire

image set $X$ with a fixed stride $S$. The time complexity of our inference approach is equivalent to only $\frac{1}{S^2}$ of that of MAESTER. Finally, we achieve pixel-level subcellular structure segmentation through clustering. We employ K-means (Lloyd, 1982) for clustering, with cluster centers calculated from a subset sampled from $X$, the size of the subset being one-thousandth of $X$.

## 4 EXPERIMENT

### 4.1 DATASET

Our dataset originates from an open-source data platform named OpenOrganelle (Heinrich et al., 2021). This data portal showcases numerous high-resolution cellular images captured by focused ion beam scanning electron microscopy (FIB-SEM). We select a dataset referred to as 'BetaSeg' in OpenOrganelle (Müller et al., 2021; Heinrich et al., 2021) for the training and testing of our model. This dataset comprises multiple high-resolution microscopic images of primary mouse pancreatic islet $\beta$ cells, with various subcellular structures annotated within the cells. These annotations are generated collaboratively by human labor and neural networks. A variety of subcellular structures are labeled, including centrioles, nucleus, plasma membrane, microtubules, Golgi apparatus, granules, and mitochondria. The dataset is divided into two groups based on whether the cells are treated with a high dose of glucose. For ease of comparison with previous work, we utilize only the group that involved high-dose glucose treatment. This subset includes four high-resolution single-cell 3D images, namely high_c1, high_c2, high_c3, and high_c4. Each 3D microscopic image has a Z-axis depth of 1000. We employ high_c1, high_c2, and high_c3 for model training, while high_c4 is used to evaluate the segmentation outcomes of the model.

### 4.2 TRAINING SETTINGS

First, we conduct random sampling within 3D images according to the size of the FOV, with each sample taken along a randomly selected axis from the X, Y, and Z axes. To make full use of the abundant microscopic cell images, we re-sample randomly in each training epoch. Given a large number of microscopic cell images, we can approximately consider each set of sampled images to be unique. This sampling method also helps to prevent the model from overfitting. Subsequently, we employ random flipping and random cropping for data augmentation to obtain our final training dataset. We select AdamW as the optimizer for our model and adjust the learning rate using a cosine decay schedule with a warm-up phase. Our model implementation is based on the vision transformer from the timm library and is trained on Nvidia RTX4090 graphics cards, with a maximum of 2800 training epochs. The detailed training settings are shown in Table 2.

### 4.3 MODEL SETTINGS

We test the HiS4MAE on high_c4, utilizing only the student branch of the encoder to encode high_c4 for clustering to achieve pixel-level subcellular structure segmentation. We compare HiS4MAE with several transformer-based self-supervised and supervised methods to validate its effectiveness and advancement. We compare it with MAESTER (Xie et al., 2023), SdAE-recon (SdAE (Chen et al., 2022) with an additional decoder), Vanilla ViT (Dosovitskiy et al., 2021), and Segmenter (Strudel et al., 2021). For both MAESTER and SdAE-recon, we also achieve segmentation by clustering the encoded high-dimensional vectors. We have also attempted segmentation using the original architecture of SdAE, but due to the difference in pretext tasks, its performance in segmentation

Table 1: Model parameter details

| parameter | value |
|---|---|
| $\alpha$ | $10^{-3}$ |
| $\beta$ | $2.5 \times 10^{-4}$ |
| $\gamma$ | 0.8 |
| $k$ | 512 |
| $embed\_dim$ | 192 |

Table 2: Training settings

| config | value |
|---|---|
| $lr$ | $5 \times 10^{-4}$ |
| $min\ lr$ | $8 \times 10^{-7}$ |
| $epoch$ | 2800 |
| $batch\ size$ | 256 |
| $sample\ number$ | $2 \times 10^4$ |

under the same method is very poor, hence we use SdAE-recon instead. It is worth mentioning that since BetaSeg fully annotates the subcellular structures of only one cell per image, for the supervised Vanilla ViT and Segmenter, we randomly sample the training set from the fully annotated regions of the images to ensure a fair comparison of the supervised methods.

## 4.4 EVALUATION

To facilitate comparison with previous work, we actually select only three types of organelles for evaluating segmentation effectiveness, categorizing the cell images into four types: nucleus, mitochondria, granules, and unrecognized. We assess the segmentation performance of all self-supervised and supervised methods based on the Dice Similarity Coefficient (DSC). For the segmentation results of self-supervised algorithms, we re-match the optimal predicted labels using the Hungarian algorithm (Kuhn, 1955).

Table 3: DSC comparison of different models. Two fully ground truth-trained supervised methods are distinguished by a dark background. The optimal results are presented in bold font, while the best results excluding the two fully supervised methods are highlighted in red font.

| model | epochs | nucleus | granules | mitochondria | unrecognized |
|---|---|---|---|---|---|
| SdAE-recon | 700 | 0.932 | 0.766 | 0.880 | 0.900 |
| SdAE-recon | 2800 | 0.849 | 0.609 | 0.719 | 0.854 |
| MAESTER | 2800 | 0.950 | 0.556 | 0.786 | 0.844 |
| HiS4MAE (ours) | 700 | 0.949 | 0.781 | 0.896 | 0.908 |
| HiS4MAE (ours) | 2800 | 0.971 | 0.800 | **0.909** | 0.916 |
| Vanilla ViT (supervised) | 2800 | **0.990** | **0.885** | 0.871 | **0.935** |
| Segmenter  (supervised) | 2800 | 0.989 | 0.872 | 0.876 | 0.932 |

## 4.5 ABLATION

To demonstrate the generalizability of HiS4MAE, we calculate the DSC for different models on the entire high_c4, which is a 3D image of size $1021 \times 545 \times 1082$. We present the DSC for all models in Table 3. Additionally, the segmentation results for some images are also listed in Figure 2. Since only the organelles of a single cell are annotated in the dataset, the results we present include only this particular cell. We perform sampling on high_c4 at multiple distinct depths along the Z-axis. For both HiS4MAE and SdAE-recon, we select the epoch with the highest DSC for demonstration.

**Self-distillation Architecture.** From the results in Table 3, it can be observed that compared to the MAE-based MAESTER, both the SdAE-recon and HiS4MAE, which incorporate self-distillation architectures, demonstrate superior subcellular structure segmentation performance in a shorter time. The performance of these two methods after 700 training epochs surpasses that of the fully trained MAESTER after 2800 epochs. To visually illustrate the differences among the models, we took slices from the top of the cell nucleus, the middle of the cell nucleus, and the bottom of the cell where the nucleus is absent. This also helps to demonstrate the generalization capability of different models for various cellular regions. As shown in Figure 2, MAESTER's pixel-level segmentation results for the cell nucleus contain some voids. In contrast, HiS4MAE exhibits good segmentation results for cell nuclei in different positions.

**Model Degradation and the Codebook.** We observe in our experiments that as the number of training epochs increases, the image reconstruction accuracy and the feature extraction capability of the encoder do not exhibit a strict positive correlation. This phenomenon occurs during the training process of SdAE-recon. In Figure 3, we present the changes in the loss function values used for image reconstruction and the average DSC values of four types of organelles during the training process of SdAE-recon. It can be seen that although the MSE loss for reconstructing images does not change significantly with the increase in training epochs, the segmentation performance of the model continuously degrades. We believe that this phenomenon is unlikely to be caused by overfitting, as

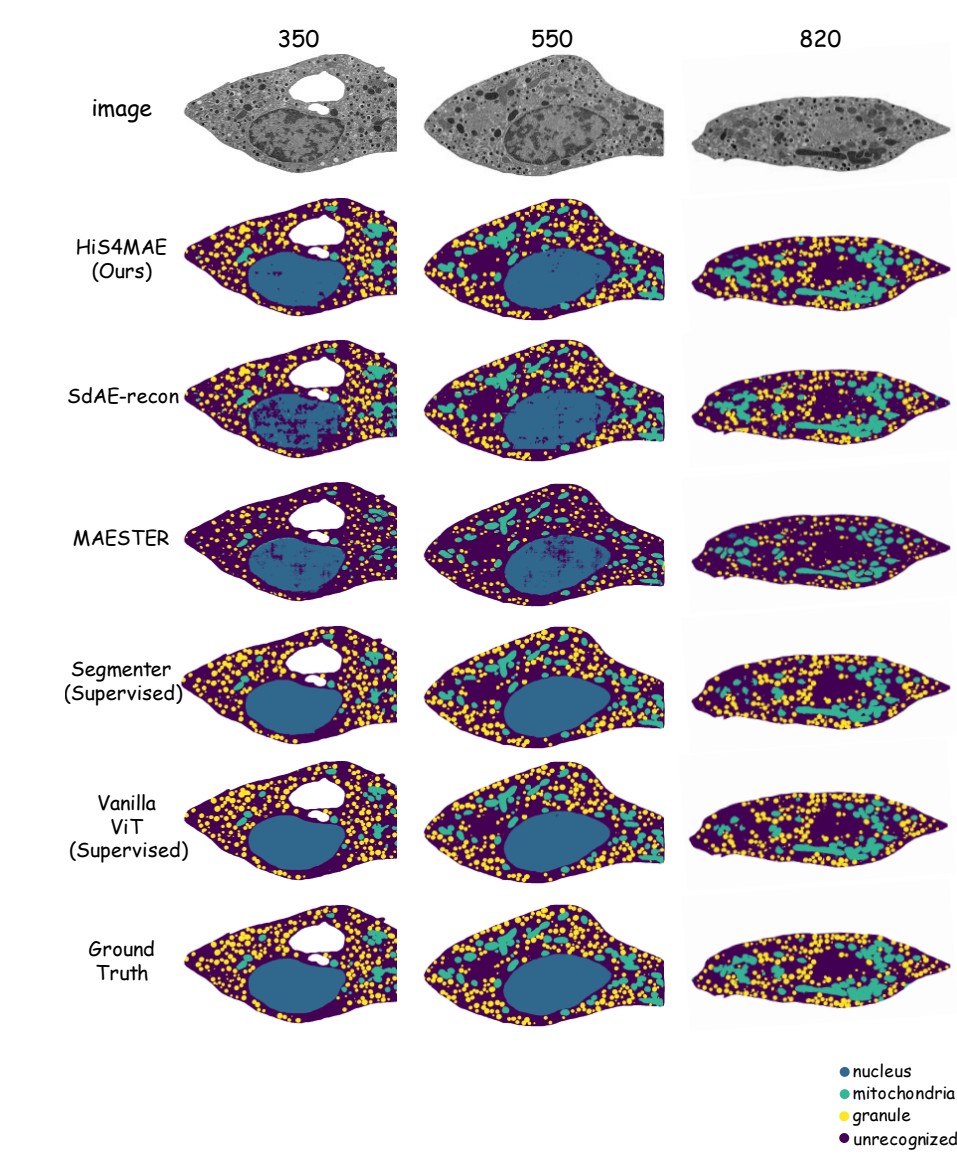

Figure 2: Segmentation outcomes across various models. The images, from left to right, represent slices of the 3D microscopic images at the 350th, 550th, and 820th layers along the Z-axis.

the model resamples the training set in each training epoch. This phenomenon results in SdAE-recon being unable to fully utilize a large number of microscopy images. As shown in Figure 2, SdAE-recon performs poorly in segmenting cell nuclei located at the top, while it performs better for cell nuclei located in the middle.

This is the reason we introduce a codebook. From Figure 3, it can be seen that after introducing the codebook, HiS4MAE exhibits more stable performance during the training process, without showing significant degradation in segmentation performance over the entire 2800-epoch training process. Moreover, HiS4MAE achieves better performance than SdAE-recon with fewer model weight parameters (by eliminating an decoder used for high-dimensional space alignment).

We attempt to explain the reasons behind this phenomenon. On one hand, the existence of the codebook allows the encoded features to aggregate around the vectors in the codebook, achieving a clustering effect similar to our inference method. On the other hand, this method acts as a special

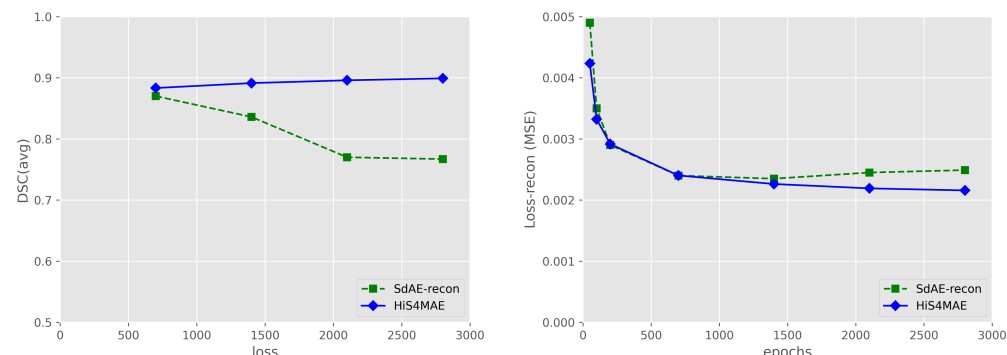

Figure 3: The variation in DSC and reconstruction loss during training for SdAE-recon and HiS4MAE. The value of DSC is taken as the average of the DSCs of the four cell structures.

form of tokenizer. Recent research (Du et al., 2024) indicates that the presence of such a tokenizer can increase the connectivity among patches that share the same vectors in the codebook within the MIM framework. Our experiments suggest that even when a tokenizer is introduced after encoding, it can improve the model's performance in downstream tasks to some extent.

**Size of the Codebook.** Our codebook is an embedding layer of size $k \times embed\_dim$. Different choices of $k$ values affect the model's performance. We fix the number of training iterations at 2800 and observe the impact of changing $k$ values on HiS4MAE. All results can be obtained from Figure 4. We can see that HiS4MAE achieves the best segmentation performance at $k = 512$, with an average DSC of 0.899. However, the model's segmentation performance does not always increase with larger $k$ values; excessively large $k$ values can slightly reduce the average DSC.

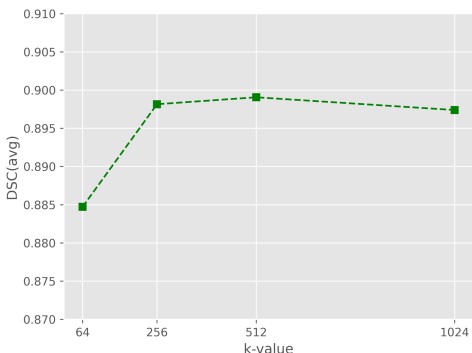

Figure 4: As the value of k increases, the segmentation results of HiS4MAE change. We evaluate the quality of the segmentation results using the average of four types of DSC.

**Visualization of Clustering Results.** We utilize UMAP (McInnes et al., 2018) to visualize the clustering results of HiS4MAE. UMAP is an algorithm that uses manifolds to reduce the dimensionality of high-dimensional data and can be employed for high-dimensional data visualization. We use UMAP to reduce the high-dimensional vectors encoded by the HiS4MAE encoder to 2 dimensions, and the results are shown in Figure 5. Generally, it is believed that the dimensionality reduction results of UMAP reflect, to some extent, the relative positional relationships of these vectors in their original dimensions. In Figure 5, the portion representing granules is closest to the unrecognized portion, and similarly, granules have the lowest DSC score among the four segmentation categories. The portion representing mitochondria is more distinct compared to granules, and thus, the DSC of mitochondria is higher. The nucleus is clustered into a separate portion far from the other or-

ganelles, and its DSC is the highest. In summary, the visualization results of UMAP also match our DSC results.

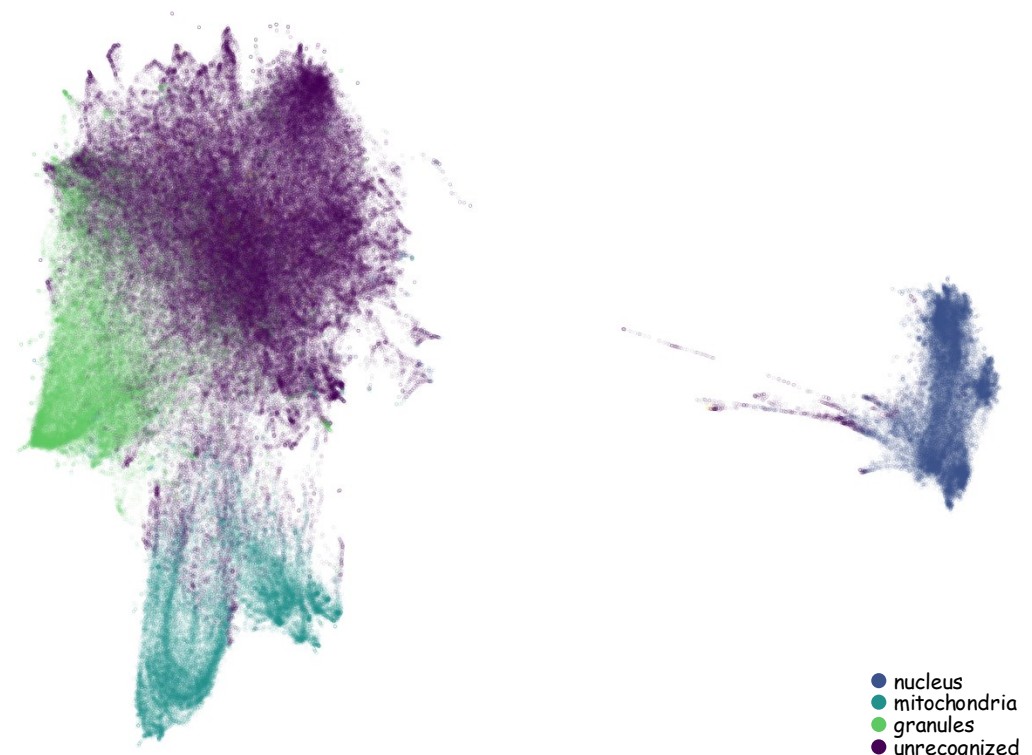

Figure 5: Visualization result of the high-dimensional vector encoded by HiS4MAE through UMAP dimensionality reduction. We label the dimensionality reduction results by UMAP with ground-truth tags.

## 5 CONCLUSION

In this work, we propose a method named HiS4MAE for subcellular structure segmentation. Our approach can fully leverage a vast array of microscopic cell images without the need for large-scale manual annotation of images. Compared to existing self-supervised methods, our technique exhibits greater stability and requires reduced training and inference times. Our proposed method not only outperforms existing state-of-the-art self-supervised approaches but also significantly narrows the performance gap with supervised methods that leverage ground truth across different categories.

It is worth further contemplation as to why merely employing a MIM approach can enable the model's encoder to achieve an effect akin to clustering on microscopic cell images. What deeper biological and mathematical principles might underlie this phenomenon? We aspire for our method and work to enhance researchers' productivity and assist them in gaining a deeper comprehension of the internal structures of cells.

## 6 ETHICS STATEMENT

We propose a masked image modeling (MIM) method based on a self-distillation architecture to address the problem of subcellular structure segmentation. Our dataset for segmentation consists of mouse pancreatic islet cell microscopy images sourced from an open-source platform. To the best of our knowledge, our research does not violate any ethical standards.

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
