# OpenReview forum: "HiS4MAE: High-efficiency Segmentation of Subcellular Structure via Self-distillated Masked Autoencoder"
_ICLR.cc/2025/Conference — Submitted to ICLR 2025_

### Official Review · Reviewer_CCuT · 2024-10-31

**Soundness:** 2
**Presentation:** 2
**Contribution:** 2
**Rating:** 3
**Confidence:** 5

**Summary:**

This paper introduces HiS4MAE, a self-supervised method for segmenting subcellular structures using a self-distillated masked autoencoder approach. The method combines an MAE architecture with self-distillation and a discrete codebook. The benchmark shows improved performance in segmenting major cell organelles.

**Strengths:**

1. To the best of the reviewer's knowledge, this is the first work to use masked image modeling with a self-distillation framework for subcellular segmentation in vEM.
2. The improvement in 4-mean clustering is substantial.

**Weaknesses:**

1. Limited evaluation scope with k-means clustering fixed at k=4:
   1. Previous works[1,2] compare the segmentation DSC on k-means centers from k=4 to k=10. Why is HIS4MAE only presenting evaluations at k=4?
   2. Subcellular structure is complex and extends far beyond just 4 major organelles. Since the authors have chosen to use k-means clustering for unsupervised segmentation, it should be straightforward to show if more clusters can correspond to finer-grained structures, e.g., the membrane sounding the granule/mitochondria. Even a qualitative segmentation map could be helpful.

2. Inconsistencies in motivations and methodologies. Line 061-063 argues that k-means' initial center introduces instability in clustering results. Line 077-078 suggests using fixed cluster centers, but line 272-273 states the method uses cluster centers calculated from random samples.

3. The expand and stride strategy is vaguely discussed. The "linear and non-linear function" on line 268 is not defined, and the strategy is not clearly explained.


[1] Han, Hongqing, et al. "Self-supervised voxel-level representation rediscovers subcellular structures in volume electron microscopy." Proceedings of the IEEE/CVF Conference on Computer Vision and Pattern Recognition. 2022.
[2] Xie, Ronald, et al. "Maester: masked autoencoder guided segmentation at pixel resolution for accurate, self-supervised subcellular structure recognition." Proceedings of the IEEE/CVF Conference on Computer Vision and Pattern Recognition. 2023.

**Questions:**

1. How is positional encoding handled? Given methods with similar architecture, e.g. MAESTER, reported the positional encoding greatly impacted the performance

2. The reviewer wonders if the codebook can be leveraged for segmentation.

---

### Official Review · Reviewer_itmP · 2024-11-03

**Soundness:** 3
**Presentation:** 4
**Contribution:** 2
**Rating:** 5
**Confidence:** 4

**Summary:**

This paper introduces HiS4MAE, a self-supervised method for efficiently segmenting subcellular structures in microscopic cell images. The proposed approach addresses the challenge of accurately identifying subcellular structures without relying on extensive manually annotated data, which is often time-consuming and labor-intensive to obtain. The authors also propose an inference method called expand-stride, which reduces the time complexity of the segmentation process compared to the cover-stride method used in the state-of-the-art approach.

**Strengths:**

1. The paper proposes a self-supervised method, HiS4MAE, for subcellular structure segmentation in microscopic cell images. The approach leverages a masked autoencoder with self-distillation and a discrete codebook to improve segmentation performance and stability.
2. The experimental results demonstrate that HiS4MAE outperforms existing state-of-the-art self-supervised methods and significantly reduces the performance gap with fully-supervised approaches.

**Weaknesses:**

1. The paper does not clearly explain how the proposed method, particularly the self-distillation architecture and the discrete codebook, contributes to the improved segmentation performance. A more in-depth analysis and discussion of the underlying mechanisms would strengthen the paper.
2. The evaluation is limited to a single dataset. To demonstrate the proposed method's generalizability, it would be beneficial to evaluate HiS4MAE on multiple datasets with a wider range of subcellular structures.
3. The paper does not provide a comprehensive comparison with other state-of-the-art self-supervised methods for subcellular structure segmentation. Comparing HiS4MAE with a broader range of relevant approaches would help to better position the proposed method in the context of existing research.

**Questions:**

1. How does the self-distillation architecture contribute to the improved segmentation performance? What are the underlying mechanisms that make it effective for this task?
2. Can the authors provide more insights into the role of the discrete codebook in stabilizing the model during training? How does it prevent model degradation, and what is the intuition behind its effectiveness?
3. How does the computational complexity and memory requirement of HiS4MAE compare to other state-of-the-art methods for subcellular structure segmentation? Can the authors provide a detailed analysis of the training and inference time, as well as the memory footprint of their method?

---

### Official Review · Reviewer_kzm6 · 2024-11-04

**Soundness:** 3
**Presentation:** 2
**Contribution:** 3
**Rating:** 5
**Confidence:** 3

**Summary:**

This paper presents HiS4MAE, a self-supervised method for subcellular structure semantic segmentation. The authors combine previous ideas from MAESTER and SdAE, and extend them with a discrete codebook to prevent model degradation during self-distillation. This shows improvements in segmentation quality and training time, reducing the gap between supervised and self-supervised methods on a dataset from OpenOrganelle.

**Strengths:**

- Interesting observation that image reconstruction accuracy does not strictly correlate with segmentation performance, and proposal to address the problem with a discrete codebook.
- Noticeable improvements in segmentation quality across all semantic classes.
- Ablation on the $k$ parameter.

**Weaknesses:**

- Evaluation limited to a single setting (BetaSeg in OpenOrganelle).
- No exploration of the impact of the FOV size.
- 2d approach (80^2 patches) to an inherently 3d problem.

**Questions:**

- "increases the overall segmentation effect"; this is imprecise, please mention the specific metric that is improved
- Were $\alpha$, $\beta$, and $\gamma$ tuned in any way?
- What's the motivation behind using MSE for $L_{eta}$ but cosine similarity for $L_{stu}$?
- How is $Z^{isolated}$ and $Z^{surrounded}$ defined in Eq. 6?
- How is $K$ selected when you use K-means?
- Could you please provide error bars for your results in Table 3?
- Can you comment on how your model is able to segment mitochondria better than a supervised approach?
- You mention that "SdAE-recon performs poorly in segmenting cell nuclei located at the top, while it performs better for cell nuclei located in the middle." Is there any depth-dependent contrast/texture variance that could explain this phenomenon?
- The name "expand-stride" is only present in the intro section, and never mentioned in the rest of the paper. For clarity, please include it in the intro section too?

---

### Meta-Review · Area_Chair_F8Gn · 2024-12-20

**Metareview:**

The paper describes a masked autoencoder framework for image segmentation of biological microscopy images. The reviewers appreciated the technical novelty of the HiS4MAE for image segmentation. However, the unanimously recommended rejection citing insufficient evaluation of the performance of the method, in particular comparisons to other work.

**Additional Comments On Reviewer Discussion:**

The reviewers engaged with the authors sufficiently during the discussion period. However, their concerns were not sufficiently addressed by the authors to improve the evaluation of the paper.

---

### Decision · Program_Chairs · 2025-01-22

Reject